# CAN LLMS RECONCILE KNOWLEDGE CONFLICTS IN COUNTERFACTUAL REASONING?

## ABSTRACT

Large Language Models have been shown to contain extensive world knowledge in their parameters, enabling impressive performance on many knowledge inten­sive tasks. However, when deployed in novel settings, LLMs often encounter situations where they must integrate parametric knowledge with new or unfamil­iar information. In this work, we explore whether LLMs can combine knowledge in-context with their parametric knowledge through the lens of *counterfactual rea­soning*. Through synthetic and real experiments in multi-hop reasoning problems, we show that LLMs generally struggle with counterfactual reasoning, often re­sorting to exclusively using their parametric knowledge. Moreover, we show that simple post-hoc finetuning can struggle to instill counterfactual reasoning ability – often leading to degradation in stored parametric knowledge. Ultimately, our work reveals important limitations of current LLM's abilities to re-purpose parametric knowledge in novel settings.

## 1 INTRODUCTION

Large Language Models (LLMs) internalize vast amounts of world knowledge during pretraining, enabling impressive performance on a wide range of knowledge-intensive tasks such as open-domain question answering, fact retrieval, and knowledge-base completion (Petroni et al., 2019; Liu et al., 2019; Roberts et al., 2020). Benchmarks like NaturalQuestions and HotpotQA have driven progress on recall-based and multi-hop reasoning, but they primarily evaluate a model's ability to regurgitate stored facts or compose chains of parametric knowledge without new external inputs (Yang et al., 2018; Kwiatkowski et al., 2019).

In contrast, many real-world scenarios require LLMs to integrate their pretrained knowledge with novel or hypothetical information provided at inference time. For example, consider a counterfactual query:

> *"If Paris were located in Italy, in which country would the Eiffel Tower stand?"*

Answering this correctly demands two distinct capabilities: *Contextual Override* and *Selective Re­trieval*. In order to override parametric knowledge using context, the model must temporarily sup­press its default fact that "Paris is in France" and accept the hypothetical premise. Furthermore, the model must still retrieve and leverage the association between "Eiffel Tower" and "Paris" stored in its weights, even though the location of Paris has been altered. Standard QA and multi-hop bench­marks do not explicitly test this dual requirement.

A growing body of work has examined knowledge conflicts and context-based overrides in LLMs. Studies on retrieval-augmented generation highlight how external documents can both help and confuse a model when facts disagree (Lewis et al., 2021). Recent analyses of multi-hop reasoning in synthetic settings ("grokking" transformers) show that models can learn to chain stored relations but typically without the capacity to absorb new premises on the fly (Wang et al., 2024a). Yet none of these works systematically probe *counterfactual* multi-hop reasoning, where the premise may conflict with or extend the pretrained knowledge graph.

In this paper, we fill that gap by asking: *Can modern LLMs selectively combine parametric knowl­edge with in-context counterfactual premises to answer multi-hop questions correctly?* Our key contributions are:

- **Counterfactual QA benchmarks.** We introduce both synthetic graph-based tasks (extending "Grokked" reasoning benchmarks) and real-world causal reasoning scenarios to isolate cases of (i) reinforcing, (ii) adding, (iii) contradicting, and (iv) irrelevant context relative to the pretrained knowledge graph.

- **Empirical analysis.** Through experiments on GPT-4o and other state-of-the-art models, we identify two primary failure modes: (a) *context-ignoring* (model defaults to stored facts) and (b) *context-overfitting* (model blindly follows the prompt). We quantify performance across standard, chain-of-thought (CoT), and fine-tuned prompting strategies.

- **Fine-tuning pitfalls.** We demonstrate that simple post-hoc fine-tuning on counterfactual examples often yields only marginal gains on the target tasks and can degrade performance on standard factual benchmarks by inducing unintended heuristics.

- **Practical implications.** We discuss the impact of our findings for interactive systems, retrieval-augmented pipelines, and safety-critical applications where accurate conditional reasoning under novel premises is essential.

Taken together, our results reveal a fundamental limitation of current LLMs: despite their remarkable capacity to memorize and retrieve facts, they lack robust mechanisms for *on-the-fly* modification or extension of their internal knowledge graph in response to conflicting or new information. Addressing this gap will require new modeling and training paradigms that can dynamically integrate stored and contextual knowledge without compromising either.

Figure 1: Concrete instantiation of the query. The counterfactual premise overrides Paris's country to Italy. A correct system performs *Contextual Override* and *Selective Retrieval* and answers **Italy**.

## 2 RELATED WORKS

**Multi-Hop QA**   Multi-hop question answering has been examined in a variety of prior works as a measure of knowledge manipulation ability. (Yang et al., 2018) introduces a benchmark for measuring multi-hop question-answering abilities for context-based question-answering. Allen-Zhu & Li (2024) examines knowledge manipulation in controlled settings, finding that LLMs can struggle to solve one-step knowledge manipulation tasks without COT or extensive fine-tuning. Wang et al. (2024a), on the other hand, shows that transformers can implicitly solve multi-hop queries in the grokking regime on a synthetic knowledge-graph task. Abramov et al. (2025) extends these findings to more real settings by augmenting Wikipedia data. Yang et al. (2024b) study the reliability of latent multi-hop reasoning on real-world knowledge, finding that certain types of relations are more conducive to multi-hop reasoning than others. Biran et al. (2024) studies the mechanistic implementation of multi-hop question-answering in LLMs, finding that it tends to arise when intermediate entities can be resolved in earlier layers of the model. . Unlike prior works, which primarily focus on multi-hop reasoning involving only parametric knowledge, we study cases in which an LLM may need to combine parametric and knowledge from its context.

**Knowledge Conflicts**   Large Language Models (LLMs) face challenges with *knowledge conflicts*, where external context clashes with internal parametric knowledge. Responses often prioritize context, parametric stores, or blend them for factual accuracy. Some methods enforce contextual premises (Yuan et al., 2024) or use attention pruning for context-exclusive outputs (Li et al., 2025). While useful for overriding facts, these approaches may be ill-suited for counterfactual reasoning, which requires selective retention and integration of parametric knowledge, not its wholesale dismissal. Conversely, closed-book QA relies solely on stored parametric facts (Petroni et al., 2019; Roberts et al., 2020) but cannot adapt to novel information or reason under hypothetical conditions deviating from this knowledge.

Hybrid techniques, especially retrieval-augmented generation (RAG), attempt to merge parametric memory with retrieved information. Methods like REALM (Guu et al., 2020), RAG (Lewis et al., 2021), Dense Passage Retrieval (DPR) (Karpukhin et al., 2020), and FiD (Izacard & Grave, 2020) condition generation on external evidence, while advanced strategies such as AdaCAD (Wang et al., 2024b) and CD2 (Jin et al., 2024) offer finer-grained balancing of these knowledge sources. Goyal et al. (2025) studies the dynamics of finetuning models for context reliance, finding that instruction tuning can often worsen context reliance. Although these methods enhance the incorporation of external factual information, they are generally not designed for the distinct challenges of counterfactual reasoning. Specifically, they do not enable LLMs to accept a hypothetical premise contradicting parametric facts, then selectively retrieve relevant parametric knowledge, and subsequently perform complex multi-hop reasoning based on this newly integrated understanding.

**Causal Reasoning and Counterfactuals in NLP**   Counterfactual reasoning is increasingly vital in Natural Language Processing (NLP), especially with Large Language Models (LLMs), whose causal reasoning capabilities are an active research area (Liu et al., 2024b). While some propose LLMs as a new frontier for textual causal discovery (Kiciman et al., 2024), critical views suggest they might be "causal parrots" merely echoing training data (Zečević et al., 2023). Benchmarking efforts consistently reveal LLM limitations: CLadder assesses formal causal reasoning (Jin et al., 2023), QRData highlights struggles with data-based causal tasks (Liu et al., 2024a), and Counter-Bench shows poor performance on formal counterfactual inference (Chen et al., 2025). Studies like Yamin et al. (2024) further detail these issues, identifying LLM failure modes such as over-reliance on parametric knowledge or narrative shortcuts. Despite these struggles, research actively explores using LLMs for specific NLP counterfactual tasks, like generating faithful explanations (Gat et al., 2024) or alternative textual outputs via counterfactual token generation (Chatzi et al., 2024). These endeavors underscore a critical gap: a deep understanding of **how and why** LLMs falter when integrating their vast parametric knowledge with novel, in-context counterfactual premises, particularly in multi-hop reasoning.

# 3 EXPERIMENTS ON REAL-WORLD LLMS

## 3.1 PROBLEM FORMULATION FOR A CAUSAL CASE

We start off by testing the LLM's capacity to reason about a very simple type of relationship between events – basic causality. In this example, we task the LLM with the binary question of determining if one event causes another event. Let's assume we have in the LLM's parametric (pre-training) knowledge that $X_0 \rightarrow X_1 \rightarrow X_n$ where $X_i, .., X_n$ are events that occur in the real world such as "rain falling" or "plants growing." We use the $\rightarrow$ notation to indicate direct causation. Simply asking the LLM if $X_0$ is a cause of $X_1$ would constitute a *one-hop* query, and asking the LLM if $X_0$ causes $X_2$ constitutes a *two-hop* query. We note that state of the art LLMs like GPT 4o generally succeed at these kinds of tasks. (Yang et al., 2024a) (All code and prompts for entire paper are in Appendix/Supplementary Materials).

Now suppose that we further introduce contextual information at test-time. For example, what if the LLM is told that instead of $X_0 \rightarrow X_1$, we have that $X_0 \rightarrow Y_1$ such that the LLM knows in its parametric knowledge that $Y_1 \rightarrow Y_2$. Can the LLM deduce that $X_0$ is now a cause of $Y_2$? Now we have to deal with a more complex counterfactual multi-hop scenario. In the main body of the paper, we include results from GPT-4o, Open AI's SOTA reasoning model GPT-5 (Thinking), a fine-tuned version of GPT-4o (GPT-5 is not fine-tuneable) and include results from LLama 3.1 in the Appendix. We categorize the contextual information introduced to the LLM into 4 partitions, as follows:

1. **Scenario 1 (Reinforcing Prior Knowledge):** Prompting the LLM with a relationship already present in its prior knowledge graph, thereby reinforcing an existing edge. Example: Given excessive rain causes flooding, query whether excessive rain causes infrastructure damage.

2. **Scenario 2 (Adding New Information):** Prompting the LLM with scenario-specific information necessary to answer the query, but absent from its parametric knowledge graph, akin to adding an edge. Example: Informing the LLM that excessive rain causes Timmy to eat vegetables, and querying whether excessive rain improves Timmy's health.

3. **Scenario 3 (Contradicting Prior Knowledge):** Prompting the LLM with information that strongly contradicts its existing parametric knowledge, equivalent to replacing an edge in its prior knowledge graph. Example: Informing the LLM that excessive rain causes desert expansion and querying whether excessive rain promotes cactus growth.

4. **Scenario 4 (Irrelevant Information):** Prompting the LLM with unrelated information, akin to providing an edge from a disconnected knowledge graph. Example: Informing the LLM that fatty food consumption causes heart attacks, and querying whether excessive rain leads to flooding.

## 3.2 PROMPTING METHODS

We compare three strategies: *Standard*—direct causal query; *CoT*—chain-of-thought prompting; *FT*—fine-tuning on counterfactual examples with CoT explanations (160 examples, hyperparameters in Appendix). Models are queried over either 1 or 2 counterfactual hops for simplicity.

## 3.3 RESULTS

We ask a binary cause/non-cause question (random baseline 50%). Figure 2 reports GPT-4o and GPT-5 accuracy across the 4 scenarios.

### 3.3.1 SUCCESS WHEN CONTEXT DOES NOT OPPOSE PRIOR

We see that in Scenario 1 where we reinforce prior knowledge, Figure 2a shows that standard prompting on GPT-4o with and without *CoT*, GPT-5(Thinking) and GPT-4o *FT*, the models perform very well with results ranging between 90% accuracy and perfect. These results together demonstrate that when contextual information reinforces existing knowledge, modern LLMs like GPT-4o and GPT-5 can reliably utilize their parametric knowledge without being misled by the prompt. Such robustness provides a useful baseline against which to compare the more challenging counterfactual scenarios in Scenarios 2 and 3.

### 3.3.2 FAILURE WHEN CONTEXT ADDS NEW INFORMATION OR CONTRADICTS PRIOR

In the adding-information scenario 2 plotted in Figure 2b, the non fine-tuned models show performance rating between 60 and 75% accuracy while *FT* improves this to ≈90% by reinforcing task-specific patterns. Under conflicting premises plotted in Figure 2c, performance collapses to near the 50% baseline with responses oscillating between stored and contextual facts with fine-tuning only marginally improving accuracy. As significant errors persist even with finetuning, this highlights the difficulty of overriding strong parametric priors. As such, it becomes clear that the greatest hurdle we encounter is information that conflicts with our prior. In a sense, scenario 2 , the second worst performance, where we add new information can be viewed as a weaker version of prior conflicting information. For example, if we look back at the previous example where we inform the LLM that excessive rain causes Timmy to eat vegetables, the LLM likely has a weak prior that rain does not cause kids to eat vegetables in general.

### 3.3.3 MIXED RESULTS FOR IRRELEVANT INFORMATION

In the irrelevant-information scenario 4 plotted in Figure 2d, we see strong performance across the board with finetuning on GPT 4o increasing the accuracy over standard and CoT GPT 4o prompting, while the reasoning model GPT-5 achieves near perfect accuracy. It should be noted that LLama 3.1 8B results (appendix) show finetuning reduces performance. As such, we see mixed signals in this regime, possibly owing to the large difference in parameter sizes in these models.

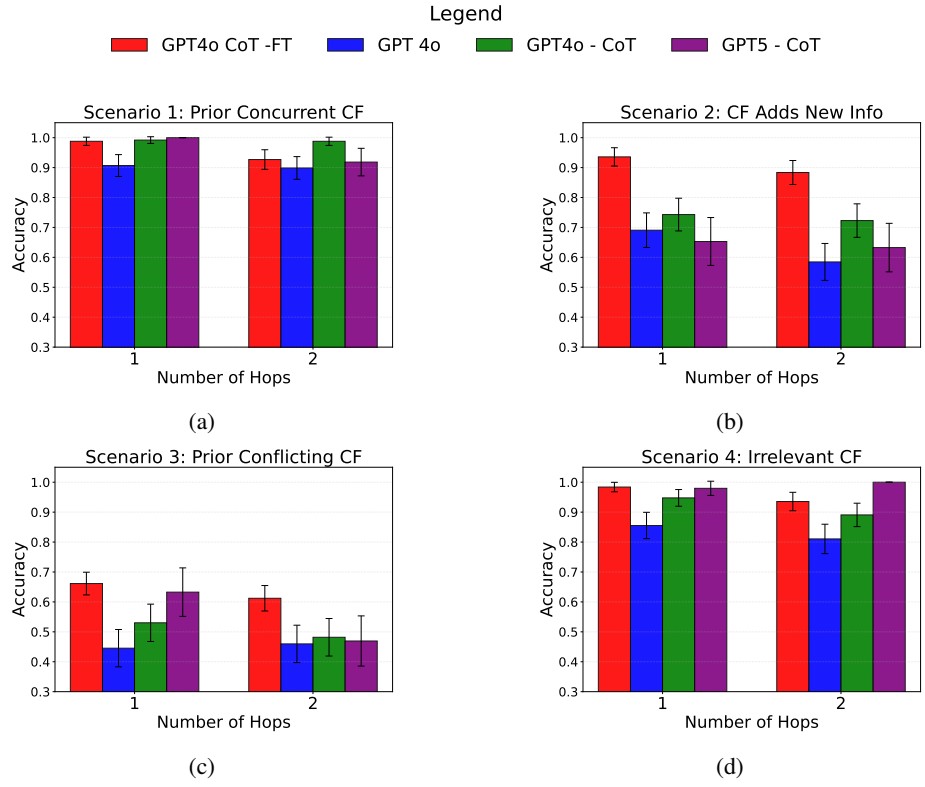

Figure 2: Causal Counterfactual (CF) Plots comparing standard GPT-4o, GPT-4o CoT, GPT-4o Fine tuned and GPT-5(Thinking) results. (a) Counterfactual Reinforces Prior, (b) Counterfactual Adds new Information, (c) Counterfactual Conflicts with Prior, (d) Counterfactual is Irrelevant to Prior and Query. 95 % CI is shown.

### 3.4 ALIGNMENT

Modern production LLMs are typically subjected to additional factuality and safety alignment stages—e.g., instruction tuning and reinforcement learning from human (or AI) feedback—which

explicitly reward consistency with memorized facts and discourage speculative deviations (Ouyang et al., 2022; Stiennon et al., 2020; Bai et al., 2022a;b; Glaese et al., 2022). This post-training pressure biases models toward relying on their pretrained parametric knowledge even when prompts present plausible counterfactual premises, making contextual overrides harder to enact reliably. This alignment likely plays a part in the results we see. To disentangle this alignment from our results, we now train our own model in the coming section.

# 4 CONCEPTUAL EXPERIMENTS IN TOY SETTING

In the previous section, we demonstrated that state-of-the-art LLMs can struggle to perform counterfactual reasoning tasks and that simple fine-tuning can be insufficient to overcome this. In this section, we perform experiments in a synthetic setup to better understand the mechanisms behind this difficulty. Our experiments take place on small transformers that we train from scratch, allowing us to study counterfactual reasoning capabilities independently of potential confounders arising from the factuality or alignment post-training of production LLMs.

## 4.1 SETUP

We perform controlled experiments in a synthetic knowledge graph setting, as used in prior studies of knowledge-based reasoning Wang et al. (2024a). To summarize, we randomly generate a directed graph $\mathcal{G}$, where each vertex represents an entity $e \in \mathcal{E}$ from a set of entities $\mathcal{E}$ where entities are linked by relations $r \in \mathcal{R}$. For simplicity, each entity and relation is identified by a unique token in the synthetic language that we study. The graph structure induces two forms of knowledge: atomic facts and inferred facts.

**Atomic facts** describe a single edge in the knowledge graph and can be expressed as a triple of $(e_i, r_j, e_k)$. These represent basic relationships between entities that must be memorized by the model.

**Inferred Facts** are those that can be deduced by a two-hop composition of the atomic facts. Inferred facts can be denoted as $(e_i, r_j, r_k, e_l)$, where there exists $e_b$ such that $(e_i, r_j, e_b), (e_b, r_k, e_l) \in \mathcal{G}$. In the inferred fact setting, we refer to the first entity, $e_i$, as the head entity and the final entity, $e_l$, as the tail. The intermediate entity, $e_b$, is referred to as the "bridge" entity. We refer to the atomic fact $(e_i, r_j, e_b)$ as the *first hop* and $(e_b, r_k, e_l)$ as the second hop. Intuitively, all atomic facts must be explicitly seen in order to specify the knowledge graph. On the other hand, a model with compositional capabilities can potentially generalize to predicting unseen inferred facts.

**Knowledge Graph Pretraining** We follow the training setup proposed in Wang et al. (2024a) to train a model that is capable of deducing unseen inferred facts by composition. Concretely, the model is presented with all atomic facts and a subset of the inferred facts. The remainder of inferred facts are reserved for the test set. We perform the pretraining for many epochs, until the model is capable of deducing the unseen inferred facts. Full knowledge pretraining hyperparameters are provided in the Appendix and largely mirror the settings reported in Wang et al. (2024a).

**Counterfactual Reasoning Task** Our primary interest in this work is the ability of LLMs to perform counterfactual reasoning over their internalized knowledge. To study this, we propose the following counterfactual reasoning task which requires the model to selectively *contextually override* its parametric knowledge with respect to certain entities, while selectively retrieving parametric knowledge relative to other entities.

We consider the setting in which the model must incorporate a new relation from the context and combine this contextual knowledge with existing parametric knowledge. Concretely, we study a setting where a modified relation, which we term as the *counterfactual premise* is input in the context, along with an inferred fact query. The LLM must then answer the inferred fact *as if* the contextually provided relation was in the knowledge graph. As inferred facts require composing two relations, successfully performing the counterfactual reasoning task requires the LLM to simultaneously override its parameteric knowledge (to incorporate the counterfactual premise) while also making use of its parametric knowledge for the remaining relation. We additionally consider cases where the

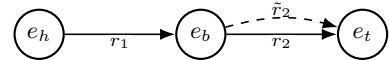

(a) **Hop-1 relevant.** CF premise $\tilde{r}_1$ overrides $r_1$; model should retrieve $r_2$ from stored knowledge.

(b) **Hop-2 relevant.** Retrieve $r_1$ from stored knowledge; CF premise $\tilde{r}_2$ overrides $r_2$

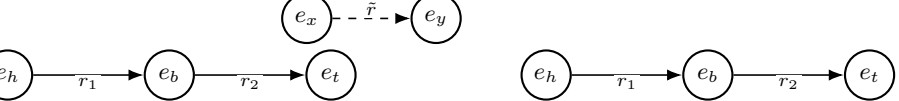

(c) **Irrelevant premise.** CF $\tilde{r}$ is unrelated; model should ignore it and use stored $r_1, r_2$.

(d) **Control (factual).** No counterfactual; require explicit two-hop trace using stored relations.

**Legend**

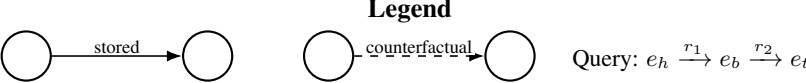

Figure 3: Conceptual visualization of toy Counterfactual (CF) tasks with shortened segments. Solid arrows are stored (parametric) edges; dashed arrows are counterfactual premises. Panels (a)–(c) are the three evaluation splits; panel (d) is the factual CoT control.

counterfactual premise is *irrelevant* to the inferred fact query (i.e. does not involve either the head or the bridge entity).

**Finetuning Stage** In our synthetic setup, the language model is only exposed to knowledge graph-consistent relations during pretraining. Mirroring our experiments on frontier model, we finetune these knowledge graph-pretrained models from the previous stage on the counterfactual reasoning task which relies partially on the pretrained parametric knowledge. As finetuning often occurs on a much narrower distribution than model pretraining, we restrict the head entities used in finetuning to come from a 20% subset of the total entities in the knowledge graph. We also balance the training and test sets equally between the following classes of examples:

1. **Hop 1 Relevant Counterfactual:** Here, the counterfactual premise modifies the first hop of the inferred fact. As an example, consider the query "If Paris was in Spain, what language would be spoken in Paris?". Here, it is discernible directly from the prompt that the counterfactual premise should is relevant to the query.

2. **Hop 2 Relevant Counterfactual:** Here the counterfactual premise modifies the link between the bridge entity and the final answer. As an example consider the query "if Paris were located in Italy, in which country would the Eiffel Tower stand?". This represents a more challenging instance of counterfactual reasoning as the model must perform the first hop before determining that the counterfactual query is relevant to the query.

3. **Irrelevant Counterfactual:** Finally, there may be cases in which the counterfactual premise is entirely irrelevant to the multi-hop query. As an example, consider the multi-hop query "If New York were located in Canada, in which country would the Eiffel Tower stand?" Here, the presence of the counterfactual premise should logically not impact the final answer.

**Incorporating Counterfactual Reasoning during Pretraining** We also examine a setting where counterfactual queries are mixed in during the pretraining stage. Concretely, we mixed in the training counterfactual prompt originally used for finetuning during the first stage of training (when atomic and memorized facts are being learned). We considered two settings: *Augmented* in which the

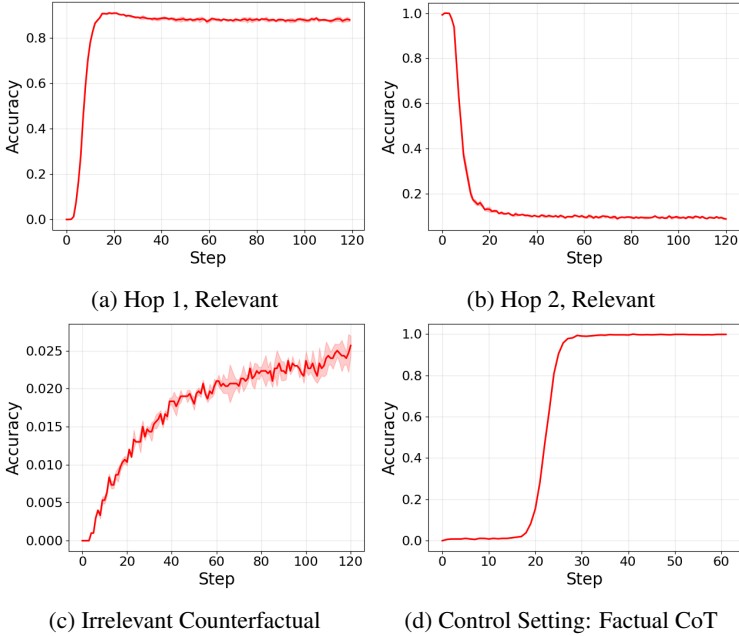

(a) Hop 1, Relevant        (b) Hop 2, Relevant

(c) Irrelevant Counterfactual      (d) Control Setting: Factual CoT

Figure 4: **Breakdown of Performance in a Conceptual Setting (a-c)** We plot the test accuracy across stages of finetuning of the three types of counterfactual reasoning queries introduced in Section 4.1. Our findings reveal that while finetuning can enable the transformer to incorporate the contextual knowledge, it is ineffective at inducing *selective* usage of contextual knowledge. As a result, performance on the *irrelevant counterfactual* split is low. **(d)** We show the performance of a factual CoT task which does not introduce any conflict with parametric knowledge. We find that fine-tuning is capable of incorporating this novel task into the model.

counterfactual examples are added in without modification and *Marked-Augmented* in which the counterfactual examples are tagged with a special token before and added in the pretraining stage.

## 4.2 FINDINGS

**Counterfactual Finetuning Induces Shortcuts** In Figure 4a,4b and 4c we show the performance of the model across the three splits throughout the finetuning process. Broadly, our results highlight that finetuning on the counterfactual reasoning task is highly susceptible to learning various short-cuts and, as a result, fails to perform uniformly well across the three splits of the evaluation dataset. Concretely, in the first stage of training, the model quickly learns to simply repeat the entity shown in the counterfactual premise. This leads to high accuracy in the *Hop 2-Relevant* split (Figure 4b) (where the final answer is always the entity present in the counterfactual premise in the query). As training continues, on the other hand, the model's performance worsens on the *Hop 2-Relevant* split , and the model transitions from performing badly to performing well on the *Hop 1-Relevant* split (Figure 4a). Notably, the model never achieves strong performance on the Irrelevant counterfactual split, despite this split being equally represented in the finetuning training set (Figure 4c). Our findings suggest a key difficulty on this task lies in learning the *selective override* mechanism: the model is capable of combining contextual and parametric knowledge, but is unable to learn to distinguish *when* the counterfactual premise is relevant.

**Performance Degradations Not a Result of Format Change** One potential explanation for the observed challenge of incorporating counterfactual reasoning ability during finetuning could arise from a more generic phenomena of *catastrophic forgetting* by which the model may forget its pre-training knowledge as it is further trained. To control for performance degradations induced by mere fine-tuning, we constructed an additional downstream task which does not require any contextual override of pre-trained knowledge. Concretely, we induce a format change where, rather than directly output the response to the inferred fact query, the model must generate a chain-of-thought

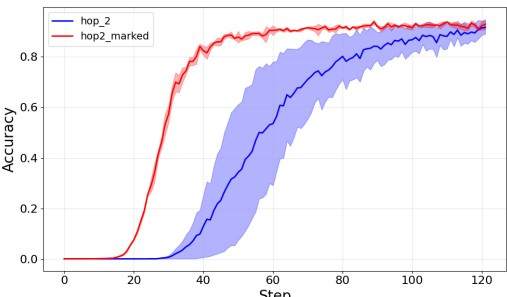

Figure 5: **Incorporating Counterfactual Data in Pretraining** We plot the worst-split accuracy across pretraining when counterfactual examples are incorporated throughout pretraining both when counterfactual examples. We observe that the in both cases, the counterfactual reasoning performance approaches $100\%$ and that marking counterfactual reasoning prompts accelerates training.

trace of the intermediate relations and entities from the head to the tail entities. As shown in Figure 4d, fine-tuning is able to quickly adapt the model to this task, as evidenced by the 100% test accuracy. This suggests that the observed failure of fine-tuning arises not from generic catastrophic forgetting, but depends on the counterfactual reasoning task. We hypothesize that the training dynamics involved in contextual override can exacerbate catastrophic forgetting.

**Effect of Incorporating Counterfactual Data in Pretraining** We explored adding counterfactual data to pretraining (augmenting 20% of KG edges, Figure 5). Incorporating counterfactual data indistinguishably from other pretraining data (*Augmented* in plot) or marking it with a special token (*Marked-Augmented* in plot) in pretraining induce good accuracies across the different groups of *counterfactual* points (i.e unrelated and related). *Marked-Augmented* performances converges faster than the *Augmented* case. It is possible that counterfactual tasks introduce interfering gradients which contribute to the suppression of parametric knowledge.

To summarize, our findings in simulation highlight that it is challenging for LLMs to *selectively override* their parametric knowledge during pretraining. In the synthetic setting, counterfactual finetuning quickly pushes the model towards over-relying on context. Notably, this contrasts with the behavior we observe in production models, where performance primarily suffers on conflicting knowledge scenarios. We attribute this difference to the fact that production models are likely to undergo significant factuality training which could bias their behavior towards consistency with parametric knowledge. Taken together our findings demonstrate that performing counterfactual reasoning tasks requires a delicate balance between preserving and overriding parametric knowledge, which is difficult to instill during finetuning.

## 5 DISCUSSION

Our work highlights LLM challenges in counterfactual reasoning requiring dynamic integration of parametric and contextual knowledge. Experiments show LLMs often default to parametric knowledge or fail to balance contextual override with selective retrieval. Simple finetuning is limited, inducing shortcuts or degrading knowledge. Pretraining with counterfactual data, while improving such reasoning, can also harm factual task performance. These findings point to a core limitation: current LLMs lack robust mechanisms for on-the-fly, conditional use of their internal knowledge. In practice, we observe two recurring failure modes—*context-ignoring* and *context-overfitting*—that persist across prompting strategies and fine-tuning. While our study sheds light on the challenges LLMs face in integrating parametric knowledge with novel counterfactual premises, it is subject to several limitations. In the toy setting, counterfactual premises are expressed as single-edge edits to a static knowledge graph and queries are limited to two-hop chains. Many real-world scenarios often involve multi-predicate interactions, ambiguous or probabilistic relationships, and noisy or conflicting evidence from multiple sources. What is significant is that we show how models struggle even under these simpler circumstances. Future work could focus on extending our analysis to deeper and more noisy multi-hop relations.

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

APPENDIX

All code is in supplementary materials

## A    CAUSAL CASE

### A.1    FULL PROMPT LIBRARY

Below are every prompt template employed in our experiments. Placeholders ('<...>') were dynamically filled in the code with the appropriate event names or lists, as described below.

**Placeholder Definitions:**

- '<Cause>': The event assumed to be the cause in the causal-query prompt (e.g., "Rain").

- '<Effect>': The event assumed to be the effect in the causal-query prompt (e.g., "Flooding").

- '<Query>': The event whose causal relationship is being queried (e.g., "Crop Growth").

- '<Target>': The event we are checking if '<Query>' causes (e.g., "Food Supply Increase").

- '<PrevEvent>': The most recently generated event in the chain when expanding synthetic event sequences.

- '<ExistingList>': A serialized list of all previously generated events in that chain (as a Python list string), used to prevent duplicates.

### A.2    A. GENERAL CAUSAL-QUERY PROMPTS

**1. Standard (Direct) Prompt**

```
Assume that <Cause> causes <Effect>.
Given this statement, does <Query> logically cause <Target>?
Only output your answer as <answer>Yes/No</answer>"
Feel free to extrapolate beyond the given context.
```

**2. Chain-of-Thought (CoT) Prompt**

```
Assume that <Cause> causes <Effect>.
Please think step by step
to decide whether <Query> logically causes <Target>.
Then output your reasoning and your answer: <answer>Yes/No</answer>
```

> **Fine-Tuned** model runs used these exact templates but pointed at the fine-tuned checkpoint.

### A.3    EVENT CHAIN GENERATION PROMPTS

**1. Causal Chain Expansion**

```
generate an event that is caused by the event: <PrevEvent>
(if this event is empty pick a random choice, do not simply
ever output the text "if this event is empty pick a random choice"),
for example Cancer -> Death or Obesity -> Bad Heart Health.
Use an arrow between the two events such that <PrevEvent> is
the first item in the chain. Make sure the event you generate
is not already in the list: <ExistingList>. Make sure the
output only includes the two events with an arrow between them.
```

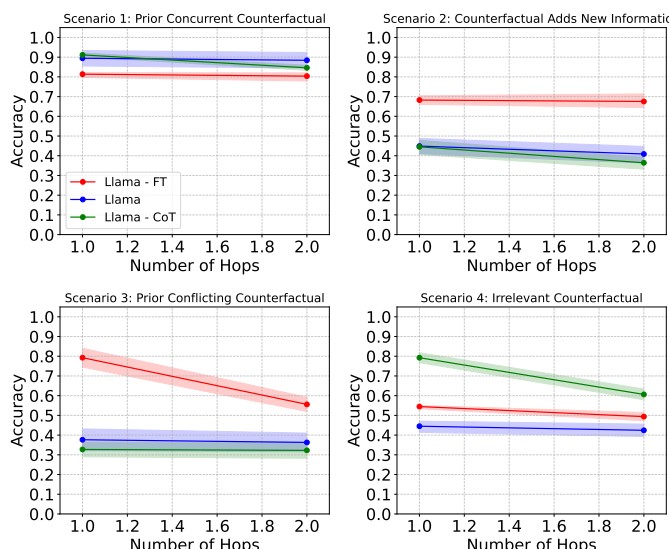

Figure 6: LLama 3.1 8B Causal Counterfactual Plots, Scenario 1 Counterfactual Reinforces Prior, Scenario 2- Counterfactual Adds new Information, (c) Scenario 3- Counterfactual Conflicts with Prior, Scenario 4- Counterfactual is Irrelevant to Prior and Query. 95 % CI is shown.

### 2. Anticausal Chain Expansion

```
generate an event that is anticausal to <PrevEvent>
(meaning the effect is actually the opposite of what it should be),
for example Cancer -> Longer Life or Obesity -> Weight Loss.
Use an arrow between the two events. Make sure the event you
generate is not already in the list: <ExistingList>. Output
only the two events with an arrow between them.
```

### 3. "Irrelevant" Transition Event

```
generate a random event that is a result of <PrevEvent>
(meaning this is a specific strange scenario), for example
Rain -> Increased Chocolate Eating or Obesity -> Warm Weather.
This event should not typically be a result of <PrevEvent>.
Use an arrow between the two events. Start with <PrevEvent>
and separate the events with an ->.
```

### 4. Post-Transition Chain Expansion

```
generate an event that is caused by the event: <PrevEvent>
(if this event is empty pick a random choice), for example
Cancer -> Death or Obesity -> Bad Heart Health.
Use an arrow between the two events such that <PrevEvent>
is the first item in the chain. Make sure the event you
generate is not already in the list: <ExistingList>.
Output only the two events with an -> between them.
```

## B  LLAMA RESULTS

In these results in Figure 6 , we see similar trends to the GPT-4o results except that we see Fine Tuning producing decreased accuracy for irrelevant counterfactuals. This mirrors what we see in the toy example.

## C GPT FINETUNING HYPERPARAMETERS

Trained tokens: 38,754 Epochs: 3 Batch size: 1 LR multiplier: 2

## D LLAMA 3.1 8B FINETUNING HYPERPARAMETERS

Epochs: 5, LoRA Rank:8,Learning Rate: 0.0001, Max Context Length:8192

## E LLM USAGE

Our paper focuses on examining the reasoning abilities of Language Models. We do not use a language model to assist with writing.

## F TOY EXPERIMENT HYPERPARAMETERS

| Hyperparameter | Values |
|---|---|
| Learning Rate | $\{10^{-5}, 5 \times 10^{-5}, 10^{-4}\}$ |
| Weight Decay | $\{10^{-1}, 10^{-2}, 10^{-3}\}$ |
| Batch Size | 512 |

## G COMPUTE

In our toy experiment, we used 4 gpus per experiment. The GPU used was NVIDIA A6000 and we had a total of 72 GPU hours.

