# OpenReview forum: "Can LLMs Reconcile Knowledge Conflicts in Counterfactual Reasoning?"
_ICLR.cc/2026/Conference — ICLR 2026 Conference Desk Rejected Submission_

### Official Review · Reviewer_fhp6 · 2025-10-14

**Soundness:** 3
**Presentation:** 2
**Contribution:** 2
**Rating:** 2
**Confidence:** 3

**Summary:**

This paper investigates the ability of LLMs to perform counterfactual multi-hop reasoning, a task requiring them to override their stored (parametric) knowledge with a new premise while selectively retrieving other relevant facts. Through experiments on both models like GPT-4o and custom-trained transformers on synthetic data, the authors find that LLMs generally fail at this task, often ignoring the context or overfitting to it. The paper argues that even post-hoc fine-tuning is an insufficient solution, often leading to performance degradation on factual benchmarks or inducing simplistic shortcuts, suggesting a fundamental limitation in current models.

**Strengths:**

1. The paper tackles an interesting problem of the reconciliation of an LLM's parametric knowledge with conflicting in-context information. It frames this challenge by "Contextual Override" and "Selective Retrieval" clearly.
2. The use of both pretrained LLMs and smaller transformers trained from scratch balances claims of real-world relevance with the controlled experiments, providing deeper insights.

**Weaknesses:**

1. The paper's central claims about model failure are difficult to verify due to a lack of clear and compelling examples. The counterfactual scenarios presented, such as the "Eiffel Tower" query, are often straightforward for state-of-the-art models, which calls into question the true difficulty of the evaluation. The argument would be stronger with more complex examples that reliably induce the claimed failures.
2. Many of the "real-world" scenarios used for testing feel contrived and lack practical utility. For instance, a premise like "excessive rain causes Timmy to eat vegetables" is quite artificial and does not reflect a plausible, real-world use case. This artificiality casts doubt on whether the observed model failures would generalize to more naturalistic and complex reasoning tasks.
3. I'm not sure if I understand correctly: in Figure 5, this result demonstrates that incorporating a relatively small amount of counterfactual data during pretraining leads to near-perfect accuracy on the task. This suggests the issue is not an inherent architectural flaw, but rather a predictable out-of-distribution generalization problem that can be rectified with appropriate training data.
4. The authors mention the introduction of a Counterfactual QA benchmark, but the paper lacks sufficient detail about it, such as dataset statistics, construction process, and representative examples.

**Questions:**

For experimental details: the hyperparameters for fine-tuning GPT model use a batch size of 1 and an LR multiplier of 2. Could you explain why using a batch size of 1? Also, what was the base learning rate to which this multiplier was applied?

---

### Official Review · Reviewer_bUz6 · 2025-10-30

**Soundness:** 2
**Presentation:** 2
**Contribution:** 2
**Rating:** 2
**Confidence:** 3

**Summary:**

The paper evaluates the ability of large language models (LLMs) to perform **two-hop counterfactual reasoning** when one hop conflicts with the model’s parametric knowledge.
For example: _“If Paris were located in Italy, in which country would the Eiffel Tower stand?”_

The authors test two main setups:
1. They compare standard prompting, CoT, and lightweight fine-tuning on GPT-4o, GPT-5 (Thinking), and an FT’d GPT-4o. They evaluate 2-hop queries in 4 categories: reinforcing existing knowledge first hop, adding new first hop, first hop contradicts priors, and irrelevant first hop.
2. A small transformer trained from scratch on factual triples, then fine-tuned on counterfactual examples. They evaluate it on 4 settings: hop 1 relevant, hop 2 relevant, irrelevant counterfactual.

Results show that LLMs generally struggle to integrate **in-context counterfactual premises** that contradict pre-training knowledge, often defaulting either to stored facts or to blindly following the context.

**Strengths:**

- **Novel benchmark.** Introduces a clear and focused benchmark for testing factual reasoning under counterfactual premises.
- **Comprehensive evaluation.** Tests multiple model types (open and closed) and different training setups.
- **Useful negative results.** The findings highlight concrete failure modes that are informative for understanding LLM limitations.

**Weaknesses:**

- **Limited evidence for claimed failure modes.**
    The paper claims that low accuracy in contradicting counterfactuals is because the model "ignores the context (defaults to stored facts)". But it’s unclear whether models actually ignore the context or partially integrate it. Qualitative error analysis (e.g., examining generated answers, attention patterns, or reasoning traces) would make the claim stronger.
- **Missing methodological details.**
    Key information is not reported: data sources, template construction, construction of the counterfactual examples, decoding settings, correctness evaluation. Without these, it’s hard to assess reproducibility and conclusions.
- **Weak motivation for 2-hop counterfactuals.**
    The paper does not convincingly argue why this setting is important or how it relates to realistic reasoning needs. Are the counterfactual for the entities used in this benchmark (Paris, Rome, etc.) something that we would actually expect models to perform well?
- **Limited contributions.**
    The benchmark and evaluations are interesting but somewhat artificial. The "failure mode" claimed is not well developed in the paper. The discussion of implications for interactive or retrieval-augmented systems is not present in the paper.

**Questions:**

1. Data and templates. What is the source of the factual and counterfactual data? Are examples derived from a knowledge graph? What relations and entities are used, and how are the textual templates constructed? How are the counterfactuals constructed?
2. Evaluation setup. Do you use log-probs or generated outputs? What are the decoding parameters? How is correctness defined (e.g. exact match) ?
3. Toy experiment. Why not use existing base models (non-RLHF) instead of training a small transformer from scratch? What is the size of the model?
4. Knowledge-graph pretraining. Are inputs textual or triplet-based? Where does this data come from, and does it match the evaluation distribution?
5. Examples and scope. Please add examples for all counterfactual types (1-hop, 2-hop, first-hop modified, second-hop modified). Why do you analyze different categories for each of the experiments, why the "hop 1 relevant" and "hop 2 relevant" only for the toy experiment and not both? why the "add new info" and others only for the first setup?
6. Figure 5. The text refers to patterns not visible in the figure. Could you clarify what is being plotted and verify consistency?

---

### Official Review · Reviewer_8ukx · 2025-10-30

**Soundness:** 1
**Presentation:** 1
**Contribution:** 1
**Rating:** 0
**Confidence:** 5

**Summary:**

This paper studies the abilities of LLMs to perform counterfactual reasoning where the counterfactual conflicts with the models parametric knowledge (knowledge formed in pretraining).
Prompting experiments based on GPT4 and 5 shows that this is an issue for state-of-the-art LLMs. And the paper also presents results for controlled finetuning and training of open source LLMs on this issue. Similar issues are found as well as how further training with the counterfacutal would help.

**Strengths:**

The paper selects an interesting point to tackle: the counterfactual reasoning abilities of LLMs.

**Weaknesses:**

I think this paper is prepared in a rush to meet the deadline, the presentation of the paper is poor, a lot of details that would help support the claims as well as help the reader understand the paper are put in the appendix.
The paper also has other weaknesses:

1. The tested models are very limited, only openai models GPT4 and 5 are tested. Therefore, we cannot make any meaningful conclusion about today's state-of-the-art models (there are other models like Gemini, Claude, Deepseek, etc), other state-of-the-art models may not have this issue. And also it is strange that no reasoning models are tested, for example, I think at least some results on openai's o1, o3, o4 models can be provided.
2. In the introduction, the problem for counterfactual is decomposed into "Contextual Override" and "Selective Retrieval" are shown, one would expect the evaluation part to include some way to quantify these two. but there's no measure for this.
3. Some examples shown are not really counterfactual, for example, Scenario 1, 2, and 4 in sec 3.1 doesn't looks like counterfactuals to me.
4. The introduction claims that one of the contributions is a new benchmark for counterfactual reasoning, but the paper is not giving any details about the benchmark (all details about the benchmark is put in the appendix), which is not a good way of organizing a paper.


I believe the paper at least need another month worth of work to meet the threshold for top conference like ICLR.

**Questions:**

See weakness

---

### Official Review · Reviewer_Dbfh · 2025-11-01

**Soundness:** 2
**Presentation:** 2
**Contribution:** 1
**Rating:** 2
**Confidence:** 2

**Summary:**

The paper aims to evaluate an LLM’s “selective retention” ability when performing counterfactual reasoning. That is, when a counterfactual query modifies a relationship, the LLM’s ability to still incorporate unmodified relationships in its reasoning. To achieve this goal, the paper introduces new benchmarks involving reasoning questions that require selective retention, both based on synthetic tasks as well as real-world scenarios.

**Strengths:**

The paper poses an interesting problem that relates a wide range research direction in language models, from multi-hop question answering to knowledge conflicts and counterfactual reasoning. The related work synthesizes all these different directions well.

**Weaknesses:**

Given the main contribution of the paper is a benchmark, the experimental evaluation is extremely limited. There are two sets of results: Section 3 involving realistic prompts and Section 4 involving a toy setting.

Experiments in Section 3 consider different models within the GPT family and consider their performance across four scenarios. As the models have different sizes, their performance with respect to each other naturally varies. The important trend is that the performance of all models drop drastically in Scenario 3, which requires selective retention. However, it is hard to determine if selective retention is the cause of this drop as there are no details about how the different scenarios are constructed, and in particular, how their difficulties are calibrated (maybe, Scenario 3 involved parametric knowledge that is overall more difficult for the LLM to recall whether it needs to be recalled selectively or not).

Experiments in Section 4 involve a synthetically generated knowledge graph, which resolves the calibration issue from the previous set of experiments. Here, authors show how the accuracy of a single LLM varies as it is being fine-tuned for counterfactual reasoning across a similar set of scenarios as the previous set of experiments. The main trend highlighted here is that there is a mode switch during training, while early on the LLM is better at “Hop 1-Relevant” questions than “Hop 2-Relevant” questions, later on it is the opposite case. While I agree with the reasoning of the authors that “Hop 1-Relevant” questions require selective retention whereas “Hop 2-Relevant” questions can be shortcutted easily, I do not understand how this mode switch supports the argument that “fine-tuning is ineffective at inducing selective retention”. At the end of fine-tuning, the accuracy of the LLM is increased in all scenarios except “Hop 2-Relevant” questions.

Overall, it is not clear to me what this new benchmark reveals what actionable insight beyond maybe that “selective retention” is a skill that current LLMs are lacking. Even for that conclusion, experiments in Section 3 do not control for confounding factors of difficulty (if they do, it is not discussed how). Meanwhile, experiments in Section 4 are set up to test a different hypothesis, whether LLMs can be fine-tuned to selectively retain. It seems like the results suggest they can be, although maybe not as effective as other tasks given the control setting in FIgure 4d.

Related work mentions how existing approaches to resolving knowledge conflicts might inadvertently hurt selective retention, which is intuitively sound. However, this is not demonstrated experimentally. This, for instance, could have been an interesting use of the benchmark.

Another minor point is that Figure 4d is a separate fine-tuning run than Figures 4a-c, which could be made more explicit by separating that run as a separate figure.

**Questions:**

In Section 3, I did not understand how 1-Hop questions work. For Scenario 3, don't questions need to involve at least two hops (one overridden, the other needs to be recalled)?

---

### Note · Program_Chairs · 2026-01-17
**Submission Desk Rejected by Program Chairs**

The following references in this submission do not refer to real documents and/or have major errors in bibliographic information:

 Xiaoyu Liu, Zuobang Ma, Jiarui Liu, Pengfei Liu, Chen Gong, Yabo Niu, Ling Geng, Yunjia Xi, Shuai Li, Emiao Lu, Sarvnaz Karimi, and Ruichu Cai. Large Language Models and Causal Inference in Collaboration: A Survey, 2024b.